# Identification and Characterization of the *HbPP2C* Gene Family and Its Expression in Response to Biotic and Abiotic Stresses in Rubber Tree

**DOI:** 10.3390/ijms242216061

**Published:** 2023-11-07

**Authors:** Qifeng Liu, Bi Qin, Dong Zhang, Xiaoyu Liang, Ye Yang, Lifeng Wang, Meng Wang, Yu Zhang

**Affiliations:** 1Sanya Institute of Breeding and Multiplication, School of Tropical Agriculture and Forestry, Hainan University, Haikou 570228, China; liuqifengedu@163.com (Q.L.); zhangdong19921@hotmail.com (D.Z.); liang2017@hainanu.edu.cn (X.L.); yyyzi@tom.com (Y.Y.); 2Key Laboratory of Biology and Genetic Resources of Rubber Tree, Ministry of Agriculture and Rural Affairs, State Key Laboratory Incubation Base for Cultivation & Physiology of Tropical Crops, Rubber Research Institute, Chinese Academy of Tropical Agricultural Sciences, Haikou 571101, China; qinbi126@163.com (B.Q.); lfwang@catas.cn (L.W.); 3Danzhou Investigation & Experiment Station of Tropical Crops, Ministry of Agriculture and Rural Affairs, Danzhou 571737, China

**Keywords:** *HbPP2C*, rubber tree, bioinformatics, expression analysis

## Abstract

Plant *PP2C* genes are crucial for various biological processes. To elucidate the potential functions of these genes in rubber tree (*Hevea brasiliensis*), we conducted a comprehensive analysis of these genes using bioinformatics methods. The 60 members of the *PP2C* family in rubber tree were identified and categorized into 13 subfamilies. The PP2C proteins were conserved across different plant species. The results revealed that the *HbPP2C* genes contained multiple elements responsive to phytohormones and stresses in their promoters, suggesting their involvement in these pathways. Expression analysis indicated that 40 *HbPP2C* genes exhibited the highest expression levels in branches and the lowest expression in latex. Additionally, the expression of A subfamily members significantly increased in response to abscisic acid, drought, and glyphosate treatments, whereas the expression of A, B, D, and F1 subfamily members notably increased under temperature stress conditions. Furthermore, the expression of A and F1 subfamily members was significantly upregulated upon powdery mildew infection, with the expression of the *HbPP2C6* gene displaying a remarkable 33-fold increase. These findings suggest that different *HbPP2C* subgroups may have distinct roles in the regulation of phytohormones and the response to abiotic and biotic stresses in rubber tree. This study provides a valuable reference for further investigations into the functions of the *HbPP2C* gene family in rubber tree.

## 1. Introduction

PP2C (protein phosphatases type 2C) enzymes constitute a monomeric group of protein phosphatases with a specific action on serine/threonine residues, and they rely on the presence of Mg^2+^ or Mn^2+^ ions for their enzymatic activity. These enzymes are highly conserved throughout evolution and are found ubiquitously among various life forms, including archaea, bacteria, fungi, plants, and animals [1,2]. Notably, plants exhibit the highest abundance of PP2C-like proteins in comparison to other organisms [3]. The typical structure of most plant PP2C proteins comprises three distinctive motifs: a conserved catalytic structural domain located at the C-terminus, an extended region housing membrane-localized signaling sequences with diverse functions at the N-terminus, and a structural domain facilitating interactions akin to receptor kinases [4].

The *PP2C* gene family exerts a significant influence on plant growth and development due to its unique structure and function. These effects encompass a wide range of processes, such as plant root formation [5,6], leaf senescence [7,8], seed germination [9], plant innate immunity [10,11], and pollen germination [12,13]. Moreover, beyond its contributions to plant growth and development, the *PP2C* family plays a pivotal role in responding to both biotic and abiotic stresses, including fungal pathogens, extreme temperatures, drought, and salt stress [2,14,15,16,17,18]. Furthermore, the *PP2C* family is integral to hormone signaling pathways, including those involving abscisic acid [7,19,20], MeJA [21], and salicylic acid [21]. Consequently, genes within the *PP2C* family are pivotal in plant responses to adverse stress conditions and the regulation of various hormone pathways.

Despite extensive research on the *PP2C* gene family in other plant species, studies on this family in rubber tree are limited. Therefore, it is imperative to identify the *PP2C* gene family in rubber tree and analyze its expression under stress conditions. In our study, we performed a comprehensive analysis of 60 *HbPP2C* genes in rubber tree, encompassing physicochemical properties, phylogenetic relationships, gene structures, conserved domains and motifs, and promoter cis-acting elements. Additionally, we investigated the expression patterns of *HbPP2C* genes in six tissues of rubber tree, their response to abiotic stress, and their expression following infestation by powdery mildew. Our findings not only serve as a reference for delving deeper into the functions of the *PP2C* gene family, but also provide a foundational understanding of the potential molecular mechanisms of *PP2C* involvement in various hormonal regulatory pathways, responses to adversity stress, and immune regulation in rubber tree.

## 2. Results

### 2.1. Characterization and Analysis of PP2C Gene Family Members in Rubber Tree

A total of 60 *HbPP2C* genes were successfully identified in the rubber tree using the genome ‘Reyan733397’ as the reference, with *Arabidopsis PP2C* genes serving as the query sequences. Key features of the *HbPP2C* genes and their encoded proteins were predicted and are presented in Table 1. These characteristics included CDS (coding sequences) length, the number of amino acids, exons, molecular weights, and PI values. Among the *HbPP2C* genes, the CDS length ranged from 681 to 3285 base pairs, and the number of exons varied from 1 to 23. The number of amino acids of the HbPP2C proteins spanned from 226 to 1094, with PI values falling within the range of 4.62 to 9.76. Moreover, the molecular weights exhibited considerable diversity, ranging from 14,735.89 Da to 57,315.53 Da. The nomenclature for these *HbPP2C* genes was based on homologs found in *A. thaliana*. Subsequently, 40 HbPP2C proteins were selected for further investigation from subgroups A to L, as illustrated in Figure 1.

**Table 1 ijms-24-16061-t001:** Characteristics of the *HbPP2C* family of genes and the corresponding proteins in rubber tree.

Gene Name	Gene ID	Length of CDS (bp)	Number of Exons	Predicted Protein		
				Siza (aa)	MW (Da)	PI
*HbPP2C1*	110650351	1236	4	411	43,539.76	5.92
*HbPP2C2*	110664056	681	4	226	25,130.49	5.41
*HbPP2C3*	110644406	1311	3	436	36,712.72	5.47
*HbPP2C4*	110640824	2001	5	666	65,573.51	5.39
*HbPP2C5*	110666131	1287	7	428	46,120.9	6.40
*HbPP2C6*	110670175	1251	4	416	45,312.93	5.21
*HbPP2C7*	110659418	1506	4	501	54,241.09	4.62
*HbPP2C8*	110653766	1086	6	361	39,711.86	6.21
*HbPP2C9*	110667550	882	5	293	26,712.08	8.31
*HbPP2C10*	110672253	846	5	281	30,766.68	8.70
*HbPP2C11*	110634536	990	8	329	30,162.58	4.79
*HbPP2C12*	110664354	1275	8	424	46,211.85	5.62
*HbPP2C13*	110651406	861	1	286	18,178.68	5.63
*HbPP2C14*	110639512	999	4	332	33,255.94	7.96
*HbPP2C15*	110666808	1539	5	512	30,078.9	8.72
*HbPP2C16*	110641533	1638	4	545	58,926.58	4.71
*HbPP2C17*	110652402	1032	10	343	39,528.75	5.39
*HbPP2C18*	110638740	1515	6	504	56,592.87	5.49
*HbPP2C19*	110651937	3285	16	1094	121,034.7	5.05
*HbPP2C20*	110665418	849	4	282	28,091.82	6.17
*HbPP2C21*	110657379	1032	9	343	31,444.84	6.09
*HbPP2C22*	110637316	1140	3	379	31,413.62	5.15
*HbPP2C23*	110659164	1170	2	389	27,827.86	4.79
*HbPP2C24*	110636987	1305	3	434	36,716.07	8.19
*HbPP2C25*	110647724	1125	4	374	40,501.11	7.06
*HbPP2C26*	110666575	915	12	304	32,822.4	5.09
*HbPP2C27*	110669236	1149	4	382	41,895.72	5.24
*HbPP2C28*	110653172	1230	2	409	14,735.89	5.79
*HbPP2C29*	110657742	2364	4	787	86,792.4	5.40
*HbPP2C30*	110639138	1158	4	385	42,130.4	4.88
*HbPP2C31*	110669048	1236	5	411	26,258.97	4.99
*HbPP2C31*	110661701	2700	3	899	87,750.35	6.81
*HbPP2C33*	110630051	1539	5	512	28,981.91	8.83
*HbPP2C34*	110639297	1074	3	357	39,813.2	6.36
*HbPP2C35*	110664833	1602	4	533	51,567.47	5.54
*HbPP2C36*	110655005	852	4	283	26,636.45	6.45
*HbPP2C37*	110643225	1278	3	425	46,328.04	5.12
*HbPP2C38*	110650239	1230	3	409	44,117.94	8.54
*HbPP2C39*	110666277	855	3	284	19,352.03	5.94
*HbPP2C40*	110631453	1593	4	530	57,315.53	5.34
*HbPP2C41*	110667158	1194	4	397	44,117.94	8.54
*HbPP2C42*	110669425	777	4	258	26,124.6	5.63
*HbPP2C43*	110646201	1134	5	377	43,622.84	7.23
*HbPP2C44*	110635324	852	8	283	32,449.25	9.76
*HbPP2C45*	110637488	912	2	303	38,238.89	5.63
*HbPP2C46*	110642221	687	4	229	27,451.31	6.30
*HbPP2C47*	110672864	1158	4	385	41,882.79	5.96
*HbPP2C48*	110651368	1158	4	385	42,961.3	8.29
*HbPP2C49*	110640918	1185	5	394	43,697.35	4.93
*HbPP2C50*	110672178	1050	4	349	40,916.22	4.85
*HbPP2C51*	110645615	3072	23	1023	110,551.67	5.44
*HbPP2C52*	110668244	948	4	315	34,981.15	4.67
*HbPP2C53*	110646945	1647	3	548	59,173.58	4.68
*HbPP2C54*	110646864	1107	4	368	40,969.80	8.44
*HbPP2C55*	110655808	1548	4	515	55,811.2	6.24
*HbPP2C56*	110642429	1632	11	543	59,251.99	4.90
*HbPP2C57*	110650667	1164	5	387	42,380.07	4.97
*HbPP2C58*	110671158	876	8	291	32,120.55	8.70
*HbPP2C59*	110657354	891	10	296	31,721.07	4.77
*HbPP2C60*	110646895	1110	3	369	40,325.37	5.23

**Figure 1 ijms-24-16061-f001:**
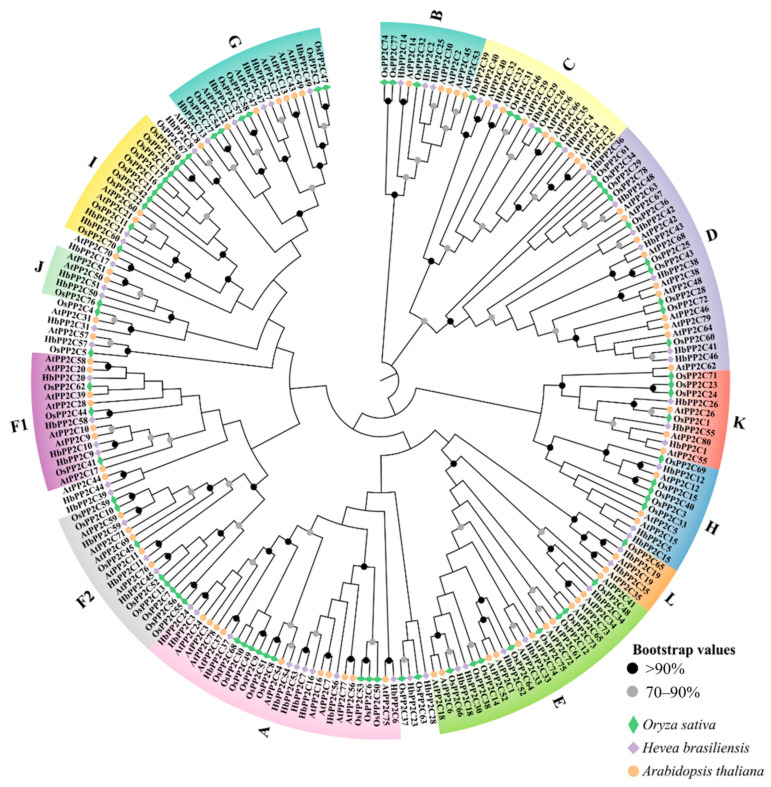
The phylogenetic tree of PP2C family proteins among 60 HbPP2Cs, 78 OsPP2Cs, and 80 AtPP2Cs. The protein sequencess were aligned by ClustalW and the unrooted phylogenetic tree constructed by the neighbor-joining (NJ) method with 1000 bootstrap replicates. The letters A to L are 13 major subfamilies of PP2Cs. Different subfamilies are represented in different colors.

### 2.2. Phylogenetic Analysis of HbPP2Cs

A comprehensive analysis of the phylogenetic relationships among the domains in HbPP2C proteins, OsPP2C proteins, and AtPP2C proteins was conducted. The PP2C proteins of three species were categorized into 13 groups, denoted as subgroups A to L and the S subgroup. The HbPP2C family consisted of varying numbers of members in each subgroup, with 9, 3, 4, 7, 5, 4, 3, 5, 3, 2, 2, 3, and 2 members in subgroups A, B, C, D, E, F1, F2, G, H, I, J, K, and L, respectively (Figure 1). Notably, subgroups A, D, and F contained five or more members. The phylogenetic tree analysis revealed a parallel evolutionary relationship between PP2C in rubber tree, *Oryza sativa*, and *Arabidopsis*. The distribution of PP2C among subgroups in rubber tree closely resembled that in *Arabidopsis* and *O. sativa*.

### 2.3. Conserved Domain and Motifs of HbPP2C Proteins in Rubber Tree

To further understand the function of HbPP2Cs, an analysis was conducted on the structure of 60 HbPP2C proteins. These proteins exhibited a common PP2Cc domain, but the positioning of this domain within the HbPP2C protein varied significantly (Figure 2). Notably, HbPP2C19 possessed the most abundant conserved structural domains, including PP2Cc, two CAP_EDs, and PKc_like domains. On the other hand, the FHA structural domain was exclusively present in HbPP2C17. In addition to identifying the HbPP2C domains, MEME website (https://meme-suite.org/meme/tools/meme, accessed on 19 May 2023) was employed to identify and map 11 conserved motifs. The majority of amino acid sequences were organized in the order of “motif8-motif1-motif5-motif4-motif2-motif6-motif3-motif7-motif10-motif9-motif11”. However, during the evolution of the PP2Cs, certain motifs appear to have been lost. All HbPP2C sequences lacked several motifs. For instance, HbPP2C26 and HbPP2C55 possessed only three conserved motifs. Most HbPP2C proteins that clustered within the same subfamily shared three or more conserved domains or motifs (Figure 2 and Appendix A). The genes within the same subgroup contained the majority of motif-encoding sequences, suggesting that these family members may have similar functions.

### 2.4. Structure of PP2C Genes in Rubber Tree

To comprehend the organization of *HbPP2C* genes, we generated diagrams illustrating the composition of their introns and exons (Figure 3). With the exception of *HbPP1C13*, which lacked introns, the *HbPP2C* gene sequences consisted of multiple introns and exons. The number of exons varied from 1 to 23, while the number of introns ranged from 1 to 22. Notably, of the 60 *HbPP2C* genes, 42 possessed between 3 and 5 exons. Among these, 23 genes contained 4 exons and 3 introns. These findings suggest a high degree of conservation in the gene structures of *HbPP2Cs*. Apart from the K, L, and J subfamily genes, the other subfamily genes displayed similar counts of exons and introns, with differences of no more than three. For instance, all members of subfamily B featured four exons, while all three members of subfamily F2 exhibited eight exons and seven introns. Moreover, *HbPP2C* genes belonging to the same subfamily in the phylogenetic tree showed comparable exon distributions and sequence lengths, indicating a certain level of conservation among genes within these subfamilies and suggesting that these genes may have similar functions.

### 2.5. Cis-Acting Element Analysis of the Promoters of the HbPP2C Genes

The promoters of all 60 *HbPP2C* genes were subjected to cis-acting element analysis. The results indicated that there were more than 38 distinct cis-elements (Figure 3 and Appendix A). These elements included five types related to plant hormones: gibberellin (P-box and GARE motif), salicylic acid (TCA element), abscisic acid (ABRE), methyl jasmonate (TGACG motif and CGTCA motif), and auxin (TGA element). This suggests that the *HbPP2C* gene family may play a role in regulating responses to phytohormones. Each promoter sequence included one to five types of plant hormone-related elements. Specifically, 42 genes had abscisic acid-responsive elements, 15 genes had auxin-responsive elements, 32 genes had gibberellin-responsive elements, 35 genes had methyl jasmonate-responsive elements, and 13 genes had salicylic acid-responsive elements (Figure 3). This indicates that a single gene may contain multiple instances of the same hormone-related element, with the highest frequency observed in the abscisic acid response element. Furthermore, these promoters also contained other important cis-elements related to responses to biotic and abiotic stresses. Examples include elements responsive to low-temperature adversity (CCGAAA), elements related to light- and drought-induced MYB binding sites (MBS), elements related to defense and stress responsiveness (TC-rich repeats), elements involved in light responsiveness (G-box), and elements related to meristem expression (CAT-box).

### 2.6. Expression Profiles of HbPP2Cs in Different Tissues

To analyze the expression of *HbPP2C* genes under different stress treatments and to explore the potential functional roles of various subfamilies, 40 *HbPP2C* genes were randomly selected for experimentation in this study. As depicted in Figure 4, *HbPP2Cs* exhibited high expression levels in branches, followed by flowers, leaves, bark, and roots, with the lowest expression observed in latex. Specifically, 29 of these genes demonstrated high expression in branches. Among these genes, subfamilies A, B, C, and D exhibited notable expression in branches. Additionally, subfamilies B, C, F1, and J displayed high expression specifically in branches and flowers. Furthermore, *HbPP2C46* and *HbPP2C23* exhibited high expression in branches, leaves, flowers, and bark. These results suggest that the *HbPP2C* genes may play essential roles in these tissues (Figure 4).

### 2.7. Expression of HbPP2C Genes under Abiotic Stress Conditions

To further investigate the role of *HbPP2Cs* in response to abiotic stress, we examined the expression of *HbPP2C* genes in rubber tree following treatments with ABA, drought, glyphosate, high temperature, and low temperature. After ABA treatment, 36 *HbPP2C* genes were upregulated in rubber tree leaves (Figure 5). Notably, members of subfamily A, specifically *HbPP2C6* and *HbPP2C53*, showed significant upregulation, with up to 31- and 25-fold increases observed at 10 and 24 h, respectively. Following drought treatment, the expression levels of 20 *HbPP2C* genes increased notably from 6 to 9 days after treatment (Figure 6). Among them, the expression levels of *HbPP2C2*, *HbPP2C25*, *HbPP2C40*, *HbPP2C29*, *HbPP2C46*, *HbPP2C23*, *HbPP2C6*, and *HbPP2C44* were strongly upregulated. The C subfamily member *HbPP2C29* exhibited the most substantial upregulation, reaching up to 33-fold. After glyphosate treatment, the expression of 29 *HbPP2C* genes remained unchanged (Figure 7). However, during 6 and 24 h of treatment, the expression of *HbPP2C54*, *HbPP2C24*, and *HbPP2C59* was significantly upregulated by up to 60-fold or more. In response to the low-temperature treatment of rubber tree, the expression trends of genes from subclades A, B, C, and D exhibited an initial upregulation, followed by a decrease and then another upregulation (Figure 8). *HbPP2C3* and *HbPP2C24* maintained high expression levels during 3–12 h of treatment, with upregulation reaching approximately 100-fold. The expression of A, B, and F1 subfamily members was upregulated from 6 to 12 h after high-temperature treatment (Figure 9). Notably, *HbPP2C9* and *HbPP2C10* were upregulated 32 times after 12 h of treatment.

**Figure 5 ijms-24-16061-f005:**
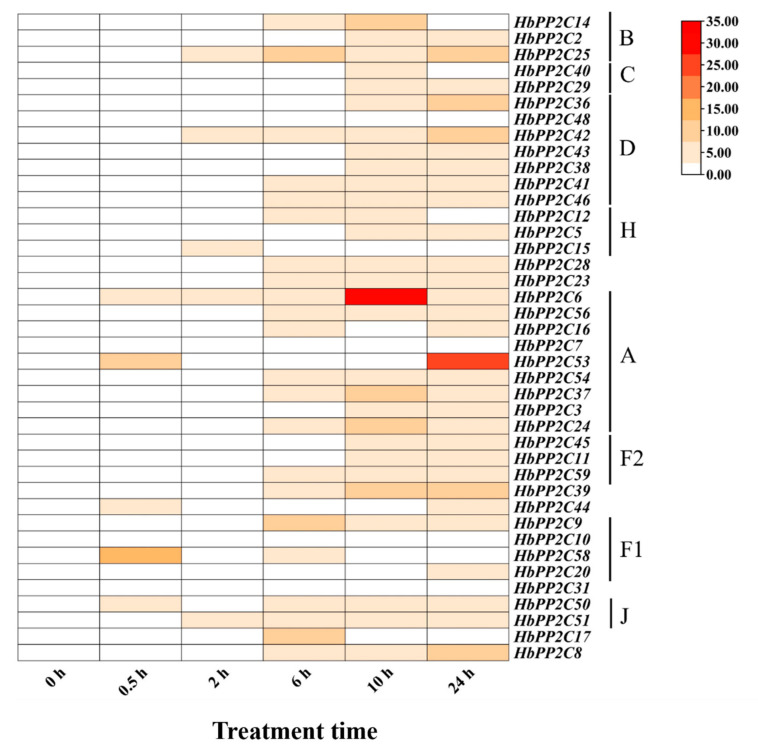
Expression of 40 *HbPP2Cs* in rubber tree subjected to ABA treatment.

**Figure 6 ijms-24-16061-f006:**
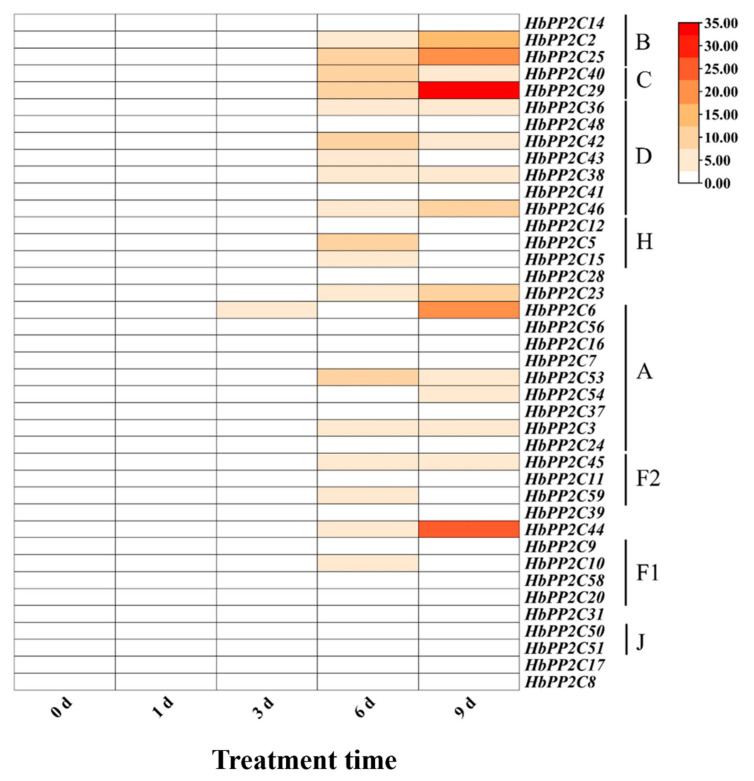
Expression analysis of 40 *HbPP2Cs* in rubber tree under drought treatment.

**Figure 7 ijms-24-16061-f007:**
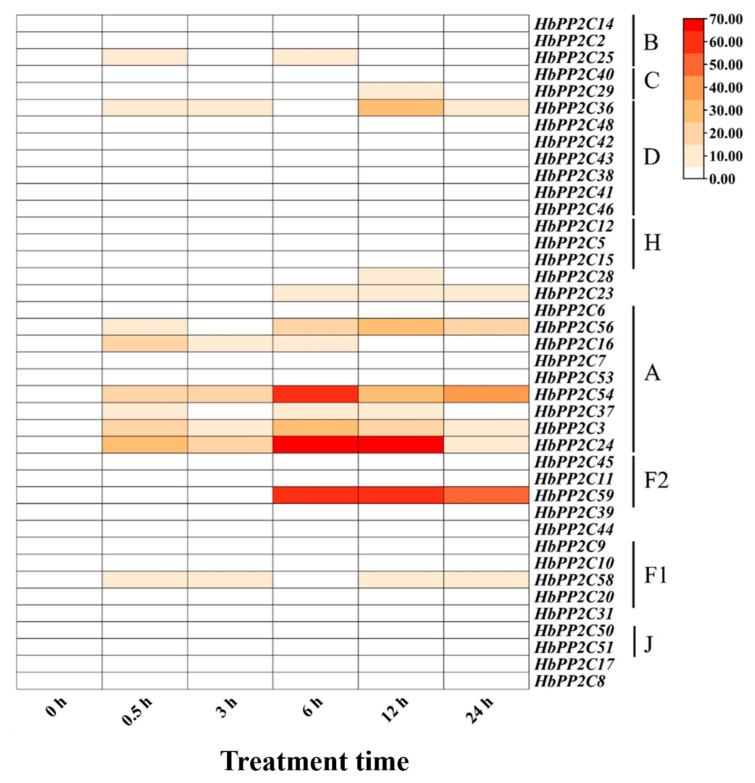
Expression patterns of 40 *HbPP2Cs* in rubber tree in response to glyphosate treatment.

**Figure 8 ijms-24-16061-f008:**
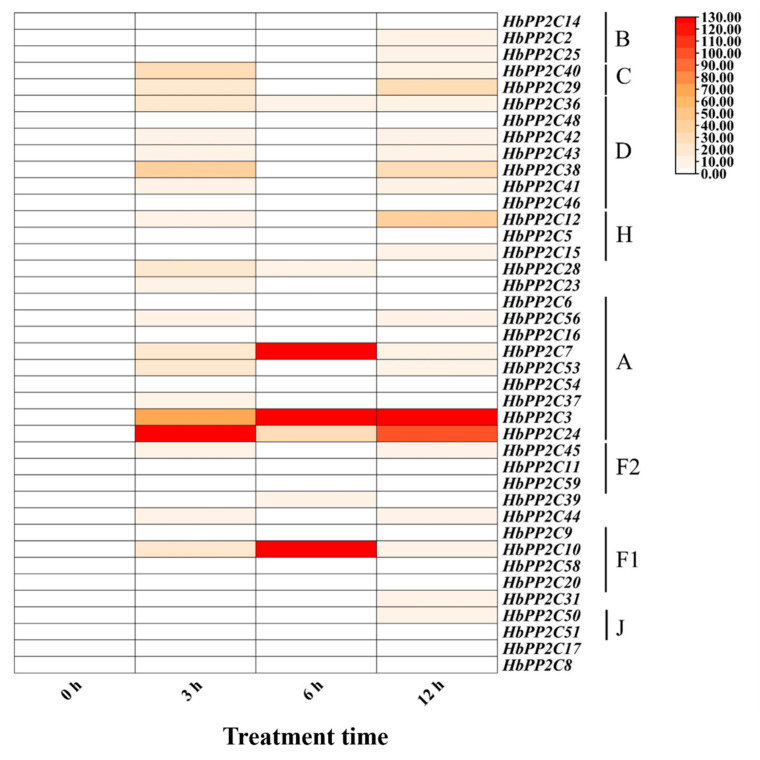
Expression analysis of 40 *HbPP2Cs* in rubber tree under low-temperature treatment.

**Figure 9 ijms-24-16061-f009:**
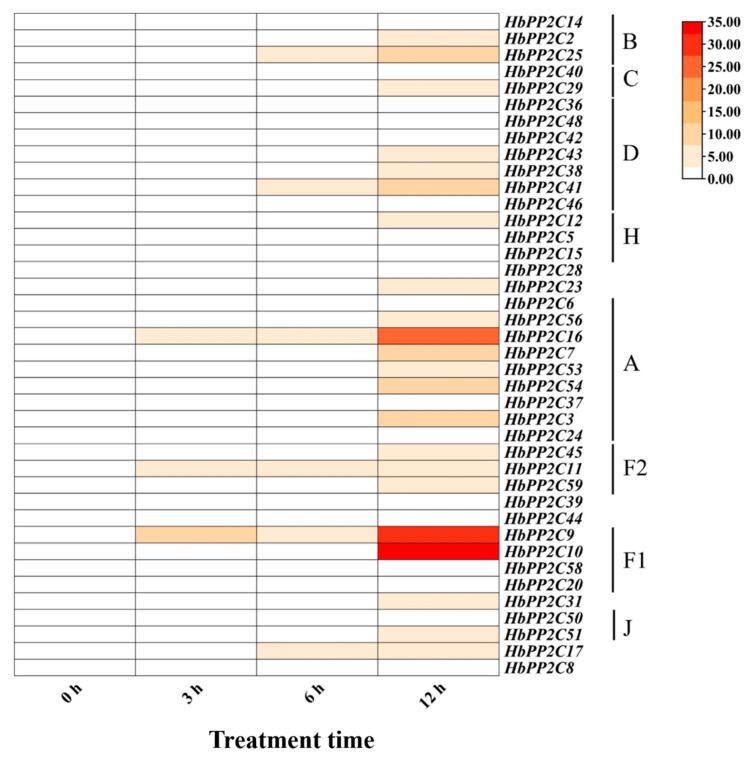
Expression patterns of 40 *HbPP2Cs* in rubber tree under high-temperature treatment.

### 2.8. Expression of the HbPP2C Gene under Biotic Stress

In response to biotic stress, we analyzed *HbPP2C* gene expression in rubber tree leaves from 0 to 24 h after powdery mildew infection. The results revealed differential expression patterns of *HbPP2Cs* following powdery mildew infection. The expression levels of all *HbPP2C* genes increased to a certain extent after 3 h of powdery mildew infection (Figure 10). Notably, genes from the A and F1 subfamilies, such as *HbPP2C6*, *HbPP2C54*, and *HbPP2C20*, exhibited significant upregulation in expression at 3, 6 and 12 h post-infection. The upregulation could exceed 10-fold, with *HbPP2C6* showing a remarkable upregulation of 33-fold, 21-fold, and 15-fold at 3, 6, and 12 months of infestation, respectively.

**Figure 10 ijms-24-16061-f010:**
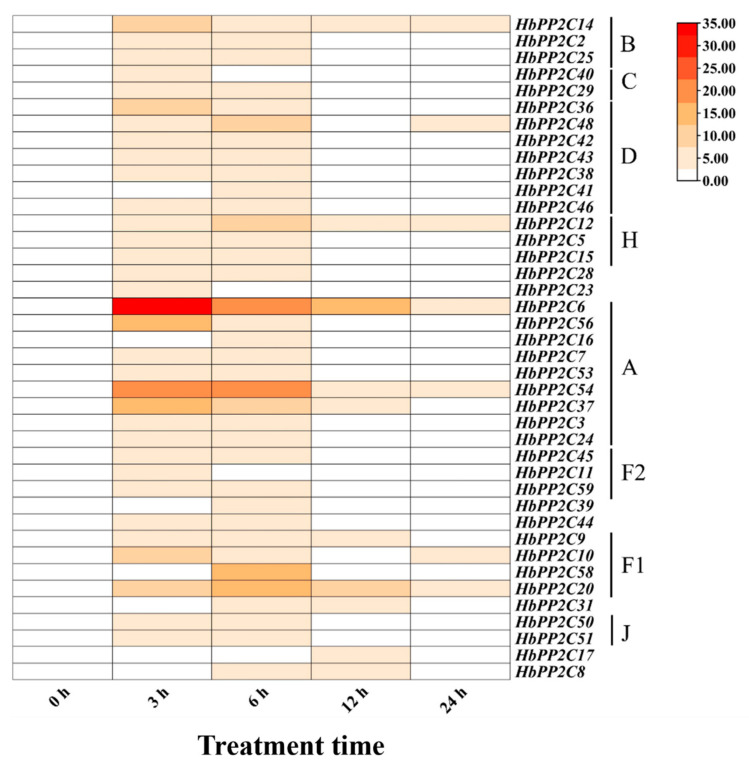
Expression analysis of 40 *HbPP2Cs* in rubber tree under powdery mildew infection.

## 3. Discussion

### 3.1. Detailed Characterization and Evolution of HbPP2Cs in Rubber Tree

Extensive research has been conducted on *PP2C* genes in model and other species, highlighting their significance. In *Arabidopsis*, a total of 80 *PP2C* genes have been identified, while in *O. sativa*, 78 *PP2C* genes have been characterized [22]. Furthermore, 209, 131, and 56 *PP2C* genes have been identified in *Brassica juncea* [13], *Brassica rapa* [23], and cucumber [3], respectively, while 62 *PP2C* genes have been found in strawberry [24]. These findings illustrate the variation in the number of PP2C family genes among various species. In this study, 60 *HbPP2C* genes were identified in rubber tree for the first time using bioinformatics methods, and the expression of 40 representative *HbPP2Cs* from different subpopulations was analyzed. To analyze the evolution of the HbPP2C family in rubber tree, an evolutionary tree was constructed, and it was discovered that the HbPP2C family exhibited similarity to that of *A. thaliana* and *O. sativa*, categorized into 13 subgroups [22] (Figure 1). The number of genes in each subfamily ranged from 2 to 9. Phylogenetic analysis showed that each subfamily of *Arabidopsis*, *O. sativa*, and rubber tree contained PP2C proteins. This indicates that certain PP2C proteins emerged prior to the divergence of monocotyledons and dicotyledons. Therefore, it can be inferred that the structure and function of HbPP2Cs are conserved during rubber tree evolution. Additionally, we noted that there were 13 K subfamily members in *Brachypodium distachyon* [25], while both rubber tree and *Arabidopsis* had only three K subfamily members. This suggests significant variability in the distribution of individual subfamily members across different species.

The diversity in exon–intron number and positional patterns often plays a significant role in the evolutionary history of gene families. In this study, we conducted an analysis of the gene structures of *HbPP2C* based on their phylogenetic relationships. The exon–intron structures showed that most members within the same subfamily shared similar exon and intron numbers but exhibited substantial differences in exon and intron lengths. Most homologous genes have highly similar amino acid sequences and exon–intron compositions. Despite the expression diversification of homologous gene pairs, their functions were preserved. It was also found that there are great disparities in the number and distribution of exons among different subfamily members. For example, the L subfamily had exon numbers ranging from 3 to 16, while the H subfamily ranged from 5 to 8 (Table 1, Figure 3). Despite the genes in these two subfamilies having close genetic relationships, there was a significant disparity in their exon–intron arrangements. All *HbPP2C* genes without introns were members of subfamily G. Genes lacking introns amplified rapidly, suggesting that in the *HbPP2C* gene family, amplification in subfamily G may be linked to gene duplication. This demonstrates that expression diversification may happen within a brief timeframe, whereas changes in functional diversity require a long period of evolution.

The catalytic structural domain of PP2C phosphatase has been shown to have eleven conserved motifs, four of which play a crucial role in maintaining Mg^2+^/Mn^2+^ homeostasis [4,26]. Our results demonstrated that HbPP2Cs within the same subfamily shared similar conserved domains and motifs. However, differences were observed between members of different subfamilies, suggesting that HbPP2Cs within the same taxonomic group share a close evolutionary relationship (Figure 3). Additionally, motif 1, motif 6, and motif 7 either significantly increased or decreased in number and shifted in position. Conversely, motif 9, motif 10, and motif 11 were lost during the evolutionary process of some PP2C proteins. For instance, members of subfamily B lost motif 9 and motif 11. These observations may be related to natural selection driving different genetic variations in rubber plants in response to various environmental challenges. This highlights the role of natural selection in shaping the genetic diversity of rubber plants in the face of adversity.

### 3.2. HbPP2C Genes Are Specifically Expressed in Branches

Previous studies showed that *PP2C* genes are differentially expressed in different tissues and can act in different tissues [27]. For example, Yang’s results showed that *MtPP2C* genes showed different expression patterns in eight tissues of *Medicago truncatula*, including roots, stems, leaves, vegetative buds, flowers, pods, and nodules. Among them, *MtPP2C2*, *MtPP2C4*, and *MtPP2C15* were highly expressed in eight tissues, while *MtPP2C34*, *MtPP2C35*, *MtPP2C36*, and other genes had low expression levels in all eight tissues. Some *MtPP2C* genes showed significantly different tissue-specific expression patterns in the eight tissues, e.g., *MtPP2C32* was significantly more highly expressed in roots than in other tissues [15]. In the present study, it was observed that the *HbPP2C* genes were most highly expressed in branches, followed by flowers, leaves, bark, and roots, and least expressed in latex. Except for *HbPP2C* gene subfamilies J and F1, all other subfamily members showed high expression in branches, demonstrating that *HbPP2Cs* may play an instrumental role in the growth and development of rubber tree. Furthermore, it should be noted that the expression patterns of *HbPP2C* genes within the same subfamily differ in different tissues. In fact, even within different tissues, the expression patterns of the same gene exhibit variations. This suggests the presence of functional diversity among them.

### 3.3. HbPP2C Genes Play Important Roles in the ABA Pathway and Drought Stress Response

Drought, as one of the most severe abiotic stresses, poses a significant threat to plant growth and crop yield [28]. Plants have evolved a series of complex signaling pathways to resist drought stress, and ABA plays a key role in this process [29,30]. The ABA signaling process is the earliest reported signaling pathway involved in *PP2Cs* and is also the most widely investigated [31]. In this study, we found that many were related to various phytohormone-associated elements and stress-responsive elements in the promoter sequence of *HbPP2Cs*. This suggests that the number and type of cis-acting elements in the promoter region of the *HbPP2C* gene may be related to its response to stress. Previous studies have shown that wheat [32], barley [17], maize [16,33,34], *O. sativa*, and *Arabidopsis PP2C* genes play important roles in plant responses to abiotic stresses, especially in ABA signaling [12,35,36,37]. In cucumber plants, the expression levels of *CsPP3C11*, *17–23*, *45*, *54*, *55*, and *2* were significantly upregulated under drought and salt treatments [3]. In *Brachypodium distachyon* [25], six genes of subfamily A were highly induced by exogenous ABA treatments, and similarly, *BdPP2C* of subfamily A was also upregulated to different degrees by drought, among which *BdPP2C36*, *BdPP2C37*, and *BdPP2C44* consistently exhibited upregulation and sustained high levels in response to all stress treatments. In addition, *BdPP2C70* of subfamily D, *BdPP2C13* of subfamily F, and *BdPP2C32* of subfamily G also showed strong expression levels under ABA treatments [25], indicating that some members of the other subfamilies may be involved and act as regulators in the ABA signaling pathway. Previous research results have shown that both glyphosate and drought can induce the regulation of ABA in plants [38]. Consistent with the results of this study, glyphosate, ABA, and drought highly induced the expression of most members of the *HbPP2C* subfamily A, B and, C genes in rubber tree at different time intervals, with a significant increase in the expression of genes. Thus, genes of subfamily A, B, and C *HbPP2Cs* may play an instrumental role in the ABA signaling pathway and drought stress response in rubber tree.

### 3.4. HbPP2C Genes Regulate the Temperature Stress Response in Rubber Tree

High- and low-temperature stress can induce *PP2C* expression in a variety of plants, such as *A. thaliana* [22], maize [39], and cotton [5]. Plants overexpressing subfamily B members of the maize *ZmPP2C* family in tobacco had higher germination rates and antioxidant enzyme activity under low-temperature stress and enhanced tolerance to cold stress, suggesting that *ZmPP2C2* may be a positive regulator of plant resistance to low-temperature stress [39]. Additionally, FGT2 is a member of subfamily D of the *Arabidopsis PP2C* family and is involved in the regulation of high-temperature stress memory by interacting with the phospholipase PLDα2 [40]. In the present study, after low-temperature treatment, each family member of *HbPP2Cs* (A, B, C, D, F1, J) showed significant upregulation at 3 h and 12 h (Figure 8). The expression levels of each *HbPP2C* family member (A, B, D, F1) were significantly increased at 12 h after high-temperature treatment (Figure 9). Hence, *HbPP2Cs* participate in the response of plants to temperature stress, and the functions of the members of subfamilies A, B, C, D, F1, and J deserve further exploration.

### 3.5. HbPP2Cs Play an Influential Role in the Response to Biological Adversity Stresses

In addition to abiotic stresses, biotic stresses such as insect pests and pathogenic bacteria also threaten plant growth. Plants have developed a series of mechanisms to resist these stresses, among which *PP2Cs* also have a positive regulatory function. Research has indicated that the expression of tomato *PP2C* genes is triggered by the fungus *Cladosporium fulvum* [41] and the oomycete *Phytophthora parasitica* [9,42]. Among the walnut (*Juglans regia*) *PP2C* family genes, *JrPP2C36* expression was significantly upregulated after inoculation with *Colletotrichum gloeosporioide* [19]. In this study, the expression levels of rubber tree *HbPP2C* genes were determined after inoculation with powdery mildew bacteria, and the expression of A (*HbPP2C6*, *HbPP2C56*, *HbPP2C54*, *HbPP2C37*) and F1 (*HbPP2C20*, *HbPP2C58*) subfamily members was significantly upregulated by up to 33-fold (Figure 10). Overall, the *HbPP2C* genes may also exert an active role when rubber tree is exposed to biotic stresses.

## 4. Materials and Methods

### 4.1. Plant Material

Seedlings and tissue samples of the rubber tree cultivar ‘Reyan 73397’ at various growth stages were sourced from the experimental base of the Rubber Research Institute, Chinese Academy of Tropical Agriculture Sciences (19°51′51 N; 109°55′63 E). Samples of 20 g roots, 20 g branches, 20 g leaves, 10 g flowers, 5 g bark and 500 mL latex of three 18-year-old healthy rubber trees were collected and used for expression analysis in different tissues. At the same time, healthy seedlings with 1 canopy of new leaves and consistent height were selected and planted in flower pots using peat soil. The seedlings were cultured with Hoagland nutrient solution every two weeks for gene expression pattern analysis.

### 4.2. Plant Treatment

Multiple groups of rubber seedlings were subjected to specific treatments, with each group consisting of three plants for experimentation. Various groups of rubber seedlings underwent specific treatments, with three plants included in each experimental group. For the hormone and chemical treatment of leaves, one group of rubber seedlings was exposed to a solution containing 200 μmol L^−1^ glyphosate and 200 μmol L^−1^ ABA. In contrast, another group of rubber seedlings received a spray of distilled water. Leaves were collected at various time points, including 0, 0.5, 2, 6, 12, 24, and 48 h post-treatment. In the drought treatment experiment, two groups of rubber seedlings were initially saturated with water in an incubator for 10 days. Subsequently, one group continued to receive regular water, serving as the control, while the other group’s water supply was discontinued. Leaves from both groups were collected at different time points: 0, 1, 3, 6, and 9 days after the cessation of watering. For the high- and low-temperature treatment experiments, rubber seedlings were placed in an incubator set at 40 °C/16 °C for the experimental group, while the control group was maintained at 25 °C. Leaves were collected from both groups at varying time intervals: 0, 3, 6, and 12 h post-treatment. In the powdery mildew infestation experiment, one group was inoculated with the *Oidium heveae* isolate HO-73, while the control group remained untreated. These rubber seedlings were then nurtured at 20 °C under conditions of 70–90% relative humidity, with a 12 h light/dark cycle. Leaves from both groups were collected at different time points: 0, 0.5, 2, 6, 12, 24, and 48 h post-treatment. Each experiment was conducted three times, and all collected samples were promptly frozen in liquid nitrogen and stored at −80 °C for subsequent analysis.

### 4.3. Gene Sequence Acquisition

The *PP2C* sequences of *Arabidopsis thaliana* and *O. sativa* were acquired from the TAIR database (https://www.arabidopsis.org/, accessed on 19 May 2023) and *O. sativa* Genome Annotation Project (http://rice.plantbiology.msu.edu/analyses_search_locus.shtml, accessed on 19 May 2023). The *A. thaliana PP2C* sequences served as templates for searching for *HbPP2C* sequences within the rubber tree genome. Sequences retrieved via BLAST were carefully curated to remove any duplicate entries. Subsequently, the open reading frames of the identified genes were searched using the NCBI ORF finder website (https://www.ncbi.nlm.nih.gov/orffinder/, accessed on 19 May 2023). Then, we employed the NCBI Conserved Domain Search to predict the amino acid sequences of these candidate *PP2C* genes.

### 4.4. HbPP2C Structural and Functional Analyses

Information regarding the *PP2C* genes, involving exon numbers and coding sequences (CDS), was sourced from the NCBI website and subsequently verified using FGENESH (http://linux1.softberry.com/berry.phtml?topic=fgenesh&group=programs&subgroup=gfind, accessed on 22 May 2023). The theoretical isoelectric point (PI) and molecular weight of the *PP2C* genes were calculated using the ProtParam website (https://web.ExPASy.org/protparam/, accessed on 25 May 2023). For further analysis, the *PP2C* gene promoter sequences (2 kb upstream of the gene initiation codon) were subjected to cis-element analysis utilizing the Plant CARE website (http://bioinformatics.psb.ugent.be/webtools/plantcare/html/, accessed on 29 May 2023) and then visualized using TBtools. Furthermore, the amino acid sequences of all 60 *PP2C* genes were used for motif prediction with the MEME Suite website (https://meme-suite.org/meme/, accessed on 22 May 2023). The results were acquired in MEME.xml format and subsequently processed for visualization using TBtools software (Version 2.003).

### 4.5. Sequence Alignment and Phylogenetic Analysis

A total of 80 *Arabidopsis* PP2C protein sequences and 78 *O. sativa* PP2C protein sequences were obtained from the TAIR database and *O. sativa* Genome Annotation Project. To analyze the obtained PP2C amino acid sequences, Multiple Sequence Alignment (MUSCLE) within Molecular Evolutionary Genetic Analysis (MEGA X) software (Version 10.0.5) was employed using default parameters. Subsequently, a phylogenetic analysis was conducted using the neighbor-joining (NJ) method, and the parameters were validated through 1000 bootstrap replicates. Additionally, a phylogenetic tree was constructed based on these sequences.

### 4.6. Expression Analysis of HbPP2Cs Using qRT–PCR

RNA extraction from all rubber tree samples followed the protocols provided by the TIANGEN Polysaccharide Polyphenol Plant Total RNA Extraction Kit (Takara, Tokyo, Japan). Subsequently, the obtained 1 µg RNA samples were subjected to reverse transcription using the RevertAid First Strand cDNA Synthesis Kit (Thermo Fisher, Beijing, China), and the resulting cDNA was utilized for qRT–PCR experiments. The primers for qRT–PCR were designed using Primer 5 (Version 5.00), with *HbActin* (GenBank accession: HO004792) serving as the internal control gene (Appendix A). The qRT-PCRs were conducted using the SYBR Premix Ex Taq II Kit (Takara, Japan) in a total volume of 20 µL. Each treatment consisted of three biological replicates per sample and three technical replicates. Expression patterns were analyzed employing the 2^−∆∆CT^ method. Statistical analysis was carried out using one-way ANOVA, and multiple comparison analyses were performed using Tukey’s test at a significance level of *p* < 0.05. The obtained results were visualized using heatmaps generated with TBtools.

## Figures and Tables

**Figure 2 ijms-24-16061-f002:**
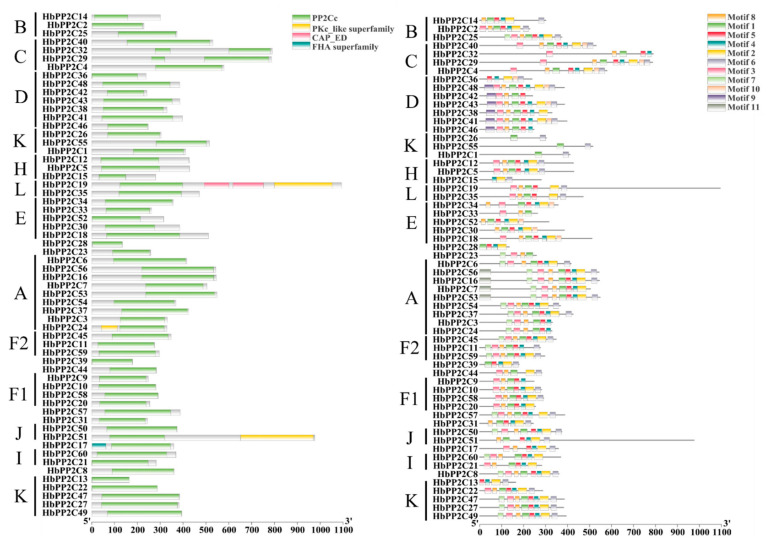
Conserved structural domains and motifs of the HbPP2C protein. The Conserved structural domains and motifs of the HbPP2C proteins were analyzed using the NCBI Conserved Domain Search and the MEME web server. Colored blocks represent conserved domains or motifs. The letters express different subfamilies of HbPP2Cs.

**Figure 3 ijms-24-16061-f003:**
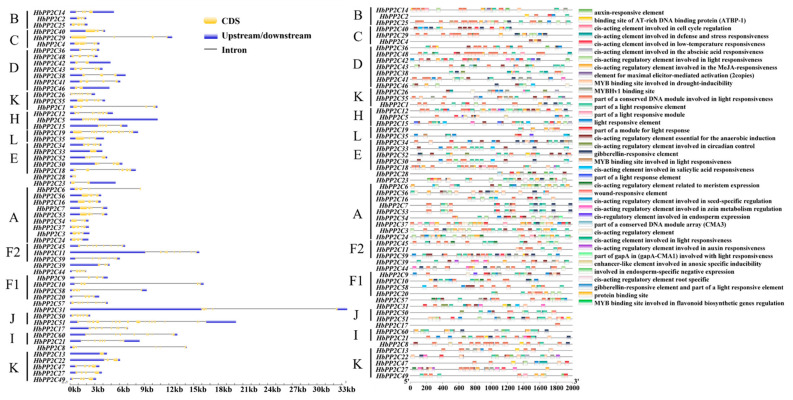
Gene structure and promoter cis-acting elements of *HbPP2Cs* in rubber tree. The letters express different subfamilies of *HbPP2Cs*.

**Figure 4 ijms-24-16061-f004:**
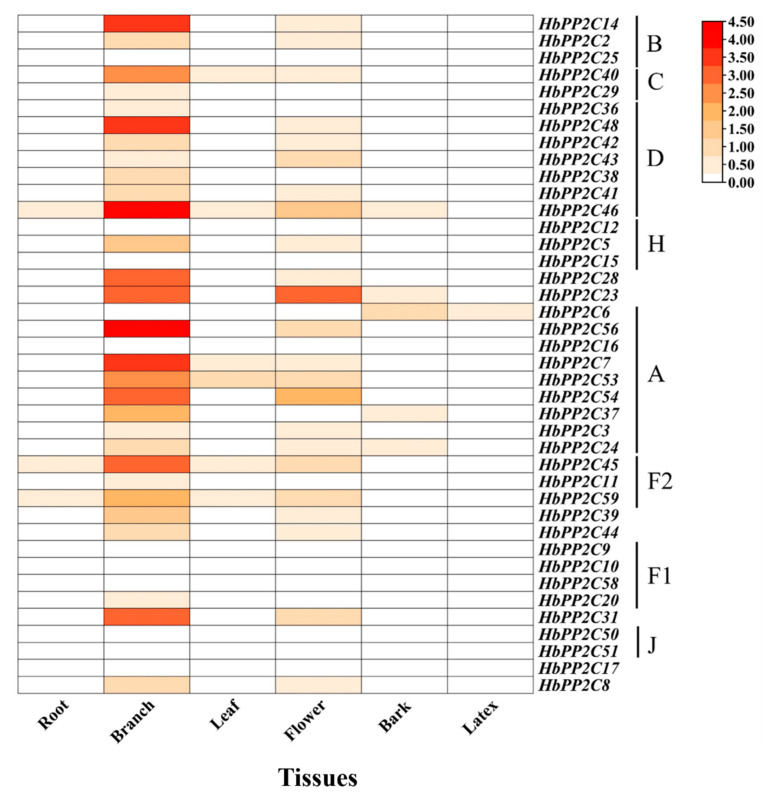
Expression profiles of 40 *HbPP2Cs* in different tissues in rubber tree. The color bar represents the gene expression level, with red indicating high expression level, white indicating no expression, and brown indicating low expression level of transcript abundance. The letters express different subfamilies of *HbPP2Cs* (the same in Figure 5, Figure 6, Figure 7, Figure 8, Figure 9 and Figure 10).

## Data Availability

All data generated or analyzed during this study are included in this article and are available upon reasonable request to the corresponding author.

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
