# Peer review of "Identification and Characterization of the HbPP2C Gene Family and Its Expression in Response to Biotic and Abiotic Stresses in Rubber Tree"

_ijms, 2023, doi:10.3390/ijms242216061_

Round 1

Reviewer 1 Report

Comments and Suggestions for Authors

The manuscript “Identification and characterization of the HbPP2C gene family and its expression in response to biotic and abiotic stresses in rubber tree” by Liu et al submitted to IJMS deals with interesting identification and expression analysis of not yet characterized gene family (HbPP2C) in ruber tree and its reactions to stresses. The manuscript will be of an interest to the scientific community working to that area and could serve as a basis for further deeper molecular studies. Below is the evaluation report.

General remark: it would be better if there are line number for easier tracking and all gene names should be in italic.

Introduction:

The introduction is well written and point the main problems related to the topic.

Materials and methods

M&Ms are somehow well written.

2.1 section should be more detailed.

2.6 section should be more detailed in terms of how many µg of total RNA was used for the analysis.

Also did the authors deposit the ruber tree sequences in public database like NCBI, if yes thishould be mentioned in the manuscript.

Results:

Results are well presented. I have a question: for the phylogenetic analysis, the authors used sequences from Arabidopsis and rice. Why didn’t they use sequences from other species especially from trees? I think it is worthy to add.

Discussion

This part is well written.

Overall, the manuscript merit to be published after addressing the above mentioned remarks.

Comments on the Quality of English Language

Author Response

Comment 1: - It would be better if there are line number for easier tracking and all gene names should be in italic?

Response: Thank you for your suggestions. The line number and gene names were modified according to the suggestions.

Comment 2: - 2.1 section should be more detailed?

Response: Section 2.1 has been modified to Section 4.1, We have incorporated your suggestions to enhance the indoor cultivation conditions and the weight of the samples for rubber tree. Please refer to line 331.

Comment 3: - 2.6 section should be more detailed in terms of how many µg of total RNA was used for the analysis. Also did the authors deposit the rubber tree sequences in public database like NCBI, if yes this should be mentioned in the manuscript?

Response: Thank you for your suggestions. Section 2.6 has been modified to section 4.6. In this experiment, 1 µg RNA was used for subsequent analysis. Additionally, the materials used in this study are consistent with the materials of the reference genome of the rubber tree. Furthermore, all primers were successfully amplified using qRT-PCR, indicating the presence of these genes. Therefore, we did not perform any additional amplification or upload the sequences to the public database.

Comment 4: - Results are well presented. I have a question: for the phylogenetic analysis, the authors used sequences from Arabidopsis and rice. Why didn’t they use sequences from other species especially from trees? I think it is worthy to add.

Response: Thank you for your suggestions. In constructing the phylogenetic tree of the developmental system, we also carefully considered this issue. Firstly, previous studies on the PP2C gene family in Arabidopsis and rice have been more in-depth and systematic, including classification and functions, etc, which has a significant guiding significance for this study. At the same time, there has been very little research on PP2C in tree species, and it is not comprehensive and in-depth enough. For these reasons, we chose the PP2C proteins from Arabidopsis and rice for the construction of the phylogenetic tree.

Reviewer 2 Report

Comments and Suggestions for Authors

Dear Editor:

Please receive my review for the manuscript entitled "Identification and characterization of the HbPP2C gene family and its expression in response to biotic and abiotic stresses in rubber tree" By Liu et al. 2023

The manuscript contains useful information on the HbPP2C genes family and their responses to biotic and abiotic stress factors in rubber tree. The manuscript generally is well written. However, it needs to be revised to respond to the below deficiencies:

1-All abbreviations need to be defined first time when they are used.

2-All figures/graphs need to have X-axis label and Y-axis label.

3-The authors need to create a section called "Experimental Design and Statistical Analysis" to explain, in details, the design of the experiment and the number of replicates involved; the statistical analysis of the data, fixed factor, random factors, and so on; and the software used. 

Author Response

Comment 1: - All abbreviations need to be defined first time when they are used.

Response: Thank you for your reminding! We have examined all the abbreviations that appear for the first time and have defined them.

Comment 2: - All figures/graphs need to have X-axis label and Y-axis label.

Response: Thank you for your suggestions. Based on your suggestions and literature review, we have made adjustments to the Figure 4-10 by adding X-axis label.

Comment 3: - The authors need to create a section called "Experimental Design and Statistical Analysis" to explain, in details, the design of the experiment and the number of replicates involved; the statistical analysis of the data, fixed factor, random factors, and so on; and the software used. 

Response: Thank you for your suggestions. The experimental design in this study can be found in Section 4.2, each experiment was conducted three times in this study, specifically on line 342. For statistical analysis, we performed variance analysis and multiple comparisons to analyze the expression trends of genes in different tissues or at different treatment times, as well as the differential expression of different genes in the same treatment. This proved the differences in gene expression. However, the heatmap cannot display the analysis process and results.

Round 2

Reviewer 2 Report

Comments and Suggestions for Authors

Dear Editor,

The authors responded to my comments. I recommend accepting the manuscript.